# Hsf1 and Hsp70 constitute a two-component feedback loop that regulates the yeast heat shock response

Joanna Krakowiak[1†], Xu Zheng[1†], Nikit Patel[2†], Zoë A Feder[1], Jayamani Anandhakumar[2‡], Kendra Valerius[1], David S Gross[3*], Ahmad S Khalil[1,4*], David Pincus[1*]

[1]Whitehead Institute for Biomedical Research, Cambridge, United States; [2]Department of Biomedical Engineering and Biological Design Center, Boston University, Boston, United States; [3]Department of Biochemistry and Molecular Biology, Louisiana State University Health Sciences Center, Shreveport, United States; [4]Wyss Institute for Biologically Inspired Engineering, Harvard University, Boston, United States

**\*For correspondence:**
dgross@lsuhsc.edu (DSG);
khalil@bu.edu (ASK);
pincus@wi.mit.edu (DP)

[†]These authors contributed equally to this work

**Present address:** [‡]Department of Biochemistry and Biophysics, Texas A&M University, College Station, United States

**Competing interests:** The authors declare that no competing interests exist.

**Abstract** Models for regulation of the eukaryotic heat shock response typically invoke a negative feedback loop consisting of the transcriptional activator Hsf1 and a molecular chaperone. Previously we identified Hsp70 as the chaperone responsible for Hsf1 repression and constructed a mathematical model that recapitulated the yeast heat shock response (Zheng et al., 2016). The model was based on two assumptions: dissociation of Hsp70 activates Hsf1, and transcriptional induction of Hsp70 deactivates Hsf1. Here we validate these assumptions. First, we severed the feedback loop by uncoupling Hsp70 expression from Hsf1 regulation. As predicted by the model, Hsf1 was unable to efficiently deactivate in the absence of Hsp70 transcriptional induction. Next, we mapped a discrete Hsp70 binding site on Hsf1 to a C-terminal segment known as conserved element 2 (CE2). In vitro, CE2 binds to Hsp70 with low affinity (9 μM), in agreement with model requirements. In cells, removal of CE2 resulted in increased basal Hsf1 activity and delayed deactivation during heat shock, while tandem repeats of CE2 sped up Hsf1 deactivation. Finally, we uncovered a role for the N-terminal domain of Hsf1 in negatively regulating DNA binding. These results reveal the quantitative control mechanisms underlying the heat shock response.
DOI: https://doi.org/10.7554/eLife.31668.001

## Introduction

The heat shock response is a transcriptional program conserved in eukaryotes from yeast to humans in which genes encoding molecular chaperones and other components of the protein homeostasis (proteostasis) machinery are activated to counteract proteotoxic stress (*Anckar and Sistonen, 2011*; *Richter et al., 2010*). The master transcriptional regulator of the heat shock response, Heat shock factor 1 (Hsf1), binds as a trimer to its cognate DNA motif – the heat shock element (HSE) – in the promoters and enhancers of its target genes (*Gross et al., 1990*; *Hentze et al., 2016*; *Sorger and Nelson, 1989*; *Xiao et al., 1991*).

In yeast, Hsf1 is essential under all conditions because it is required to drive the high level of basal chaperone expression needed to sustain growth (*McDaniel et al., 1989*; *Solís et al., 2016*). Mammalian Hsf1 is dispensable under non-heat shock conditions because it exclusively controls stress-inducible expression of its target regulon, while high-level basal chaperone expression is Hsf1-independent (*Mahat et al., 2016*). Hsf1 has been shown to play pro-cancer roles both in tumor cells and the supporting stroma (*Dai et al., 2012*; *Dai et al., 2007*; *Santagata et al., 2011*; *Scherz-*

*Shouval et al., 2014*). In addition to supplying high levels of chaperones to cancer cells, Hsf1 takes on specialized transcriptional roles to support malignant growth, and its activity is associated with poor prognosis in a range of human cancers (*Mendillo et al., 2012*; *Santagata et al., 2011*; *Scherz-Shouval et al., 2014*). Conversely, lack of Hsf1 activity has been proposed to contribute to the development of neurodegenerative diseases associated with protein aggregates (*Gomez-Pastor et al., 2017*; *Neef et al., 2011*). Despite the potential therapeutic benefits of modulating Hsf1 activity, a quantitative description of the regulatory mechanisms that control its activity in any cell type remains lacking.

Phosphorylation, SUMOylation, acetylation, chaperone binding (Hsp40, Hsp70, Hsp90 and/or TRiC/CCT), intrinsic thermosensing and an RNA aptamer have all been suggested to regulate Hsf1 in various model systems (*Anckar and Sistonen, 2011*; *Baler et al., 1993*; *Cotto et al., 1996*; *Hentze et al., 2016*; *Hietakangas et al., 2003*; *Holmberg et al., 2001*; *Kline and Morimoto, 1997*; *Neef et al., 2014*; *Shamovsky et al., 2006*; *Shi et al., 1998*; *Westerheide et al., 2009*; *Xia et al., 1998*; *Zheng et al., 2016*; *Zhong et al., 1998*; *Zou et al., 1998*). These diverse mechanisms can operate on Hsf1 by impinging on a number of steps required for activation including nuclear localization, trimerization, DNA binding and recruitment of the transcriptional machinery. Our recent work in *Saccharomyces cerevisiae* demonstrated that binding and dissociation of the chaperone Hsp70 is the primary ON/OFF switch for Hsf1, while phosphorylation is dispensable for activation but serves to amplify the transcriptional output (*Zheng et al., 2016*).

Based on these results, we generated a mathematical model of the yeast heat shock response. Given that we observed heat shock-dependent dissociation of Hsp70 from Hsf1, and that the genes encoding Hsp70 are major targets of Hsf1, we centered the model on a simple feedback loop in which Hsf1 activates expression of Hsp70, which in turn represses Hsf1 activity. While the model was able to recapitulate experimental data of Hsf1 activity during heat shock and correctly predicted the outcome of defined perturbations, its two central tenets remain untested: (1) Hsp70 directly binds to Hsf1 at a specific regulatory site; (2) Transcriptional induction of Hsp70 provides negative feedback required to deactivate Hsf1. Here, we provide direct evidence supporting these core model assumptions by severing the transcriptional feedback loop, rendering Hsf1 unable to deactivate, and mapping a direct Hsp70 binding site on Hsf1 through which Hsp70 represses its potent C-terminal transactivation domain. These results suggest that the heat shock response circuitry in this model system can be abstracted to a simple two-component feedback loop.

## Results

### Hsp70-mediated negative feedback is required to deactivate Hsf1

Our model of the heat shock response is centered on a feedback loop in which Hsf1 regulates expression of its negative modulator, Hsp70 (*Figure 1A*). When the temperature is raised, the concentration of unfolded proteins exceeds the capacity of Hsp70. Hsp70 is titrated away from Hsf1, freeing Hsf1 to induce more Hsp70. Once sufficient Hsp70 has been produced to restore proteostasis, Hsp70 binds and deactivates Hsf1. In addition to producing more Hsp70, Hsf1 also induces expression of an inert YFP reporter that can be used as a proxy for Hsf1 activity. In the yeast strains used here, this YFP reporter is integrated into the genome under the control of a promoter containing four repeats of the heat shock *cis*-element (4xHSE) recognized by Hsf1 (*Zheng et al., 2016*).

To test the model, we severed the feedback loop, both computationally and experimentally, and monitored Hsf1 activity over time following a shift from 25°C to 39°C by simulating and measuring the HSE-YFP reporter. We cut the feedback loop in the mathematical model by removing the equation relating the production of Hsp70 to the concentration of free Hsf1 without changing any parameters or initial conditions. In the absence of Hsf1-dependent transcription of Hsp70, the model predicted that the HSE-YFP reporter should be activated with the same kinetics as that of the wild type, but should continue to accumulate long after the response is attenuated in the wild type (*Figure 1B*).

To experimentally test this in yeast cells, we decoupled expression of all four cytosolic Hsp70 paralogs (*SSA1/2/3/4*) from Hsf1 regulation while maintaining the expression of total Hsp70 near its

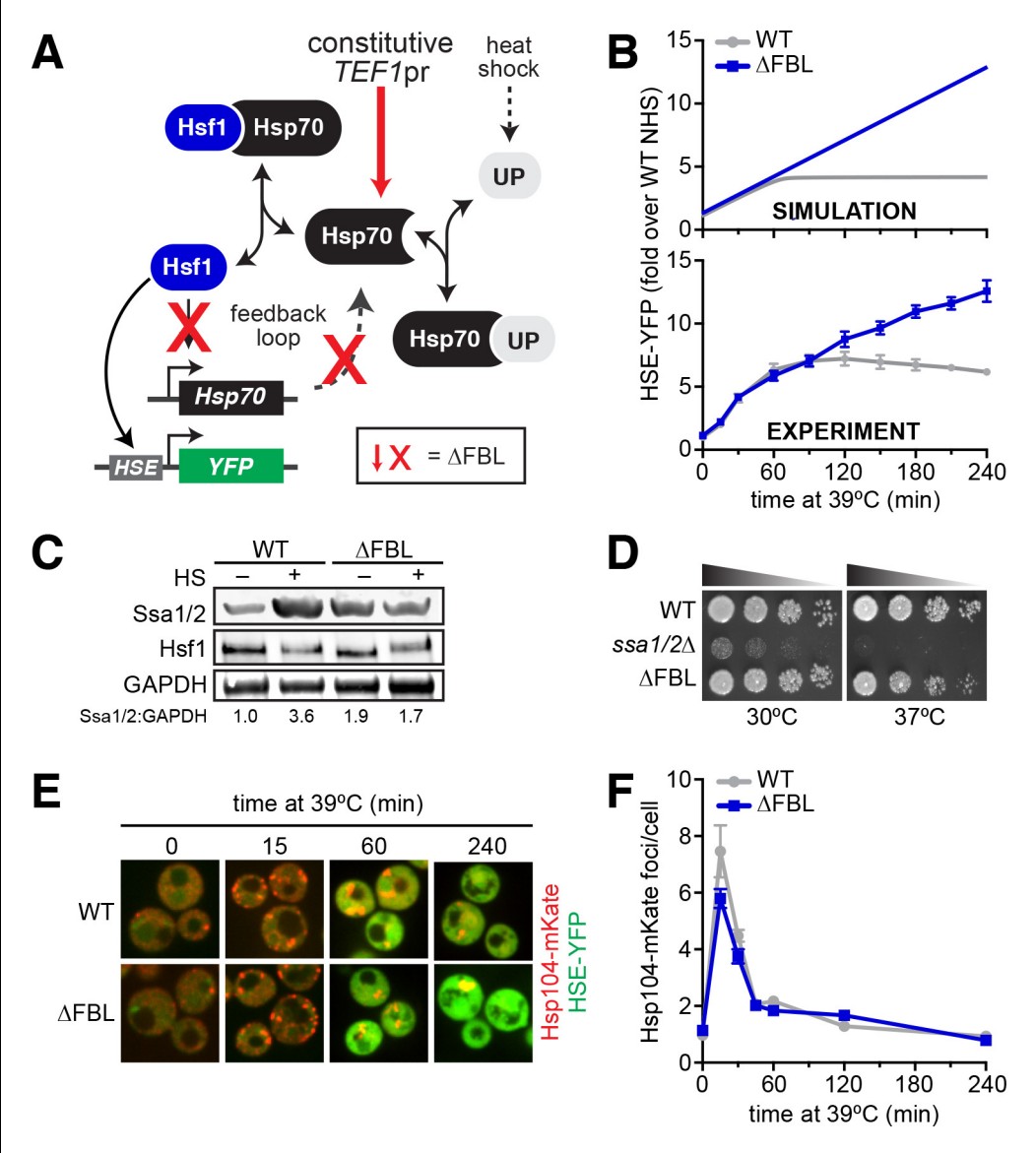

**Figure 1.** Transcriptional induction of Hsp70 during heat shock is required for Hsf1 deactivation but not proteostasis. (**A**) The Hsf1 regulatory circuit described by the mathematical model. To generate the feedback-severed yeast strain (ΔFBL), all four Hsp70 paralogs (*SSA1/2/3/4*) were deleted from the genome and 2 copies of *SSA2* under the control of the Hsf1-independent TEF1 promoter were integrated to achieve comparable Hsp70 expression under basal conditions. (**B**) Simulated and experimental heat shock time courses comparing the HSE-YFP reporter in wild type and ΔFBL cells. The experimental points represent the average of the median HSE-YFP level in three biological replicates, and the error bars are the standard deviation of the replicates. (**C**) Western blot of the expression of Hsp70 (Ssa1/2), the Hsf1 and glycolytic enzyme GAPDH (Tdh1/2/3) in wild type and ΔFBL cells under non-heat shock and 60 min heat shock conditions. The values for the ratio of Ssa1/2:GAPDH are the average of two biological replicates. (**D**) Dilution series spot assay of wild type, *ssa1/2Δ* and ΔFBL cells grown at 30°C and 37°C for 36 hr. (**E**) Wild type and ΔFBL cells expressing the Hsp104-mKate aggregation reporter along with the HSE-YFP imaged over a heat shock time course showing transient accumulation of Hsp104 foci and sustained induction of HSE-YFP levels in the ΔFBL cells. (**F**) Quantification of the number of Hsp104-mKate foci in wild type and ΔFBL cells over a heat shock time course. N > 100 cells for each time point. Error bars represent the standard error of the mean.

DOI: https://doi.org/10.7554/eLife.31668.002

The following figure supplement is available for figure 1:

*Figure 1 continued*

**Figure supplement 1.** Comparison of heat shock time course simulations of the wild type and ΔFBL mutant using the original and updated values for the parameter describing the strength of the transcriptional induction (β).
DOI: https://doi.org/10.7554/eLife.31668.003

endogenous levels under non-heat shock conditions. This was achieved by integrating two copies of *SSA2* under the control of the Hsf1-independent *TEF1* promoter into the genome and deleting *ssa1/2/3/4*. We named this strain ΔFBL to denote that we had removed the feedback loop (*Figure 1A*). As expected, wild type cells were able to increase Hsp70 levels during heat shock. By contrast, ΔFBL cells were unable to induce Hsp70 during heat shock, though the Hsp70 level was constitutively elevated (*Figure 1C*). We performed a heat shock time course in WT and ΔFBL cells and compared the expression of the HSE-YFP reporter by flow cytometry. As predicted by the simulation, the ΔFBL strain activated the reporter with identical kinetics to the wild type during the early phase of the response, but failed to attenuate induction during prolonged exposure to elevated temperature (*Figure 1B*). While the simulation correctly predicted the experimental results qualitatively, the model underestimated the amount of time required to observe the separation between the wild type and ΔFBL strains, suggesting the strength of the feedback had been exaggerated in the first iteration of the model (*Figure 1—figure supplement 1*). By reducing the strength of the feedback loop, we were able to quantitatively match the behavior of both the wild type and ΔFBL cells (*Figure 1B*, see Materials and methods for the updated parameter value).

The inability of Hsf1 to deactivate in the ΔFBL strain could result either from a specific disruption of the 'OFF switch' or from a general failure of the cells to restore proteostasis. In other words, does cutting the feedback loop simply result in sustained stress, or is the prolonged Hsf1 activity the result of specifically breaking its deactivation mechanism? To distinguish these possibilities, we first compared growth of wild type, ΔFBL and *ssa1/2Δ* cells at 30°C and 37°C. The *ssa1/2Δ* cells – which retain viability due to Hsf1-mediated induction of *SSA3/4* – displayed severely impaired growth at 30°C and were inviable at 37°C (*Figure 1D*). By contrast, the wild type and ΔFBL strains grew equally at 30°C, and the ΔFBL strain showed only a slight reduction in growth at 37°C (*Figure 1D*). Surprisingly, removal of both *SSA3* and *SSA4* – the major heat shock-inducible Hsp70 genes in yeast – had only modest phenotypic consequences at elevated temperature. The reduced growth of the ΔFBL mutant at 37°C could be a consequence of either an inadequate or overzealous heat shock response, and does not necessarily indicate a general failure to restore proteostasis. To directly monitor the loss and restoration of proteostasis, we imaged wild type and ΔFBL cells expressing Hsp104-mKate over a heat shock time course. Hsp104 is a disaggregase that forms puncta marking protein aggregates when tagged with a fluorescent protein (*Solís et al., 2016*). Upon acute heat shock, the number of Hsp104-mKate foci spiked in both wild type and ΔFBL cells, but dissolved with the same kinetics in both strains (*Figure 1E,F*). These data suggest that the ΔFBL cells can restore proteostasis just as efficiently as wild type cells and that the prolonged Hsf1 activation in the ΔFBL cells is due to a spdeactivation defect. Since Hsp104 requires Hsp70 for efficient activity (*Seyffer et al., 2012*; *Winkler et al., 2012*), there must be sufficient Hsp70 expressed under basal conditions to allow Hsp104 to operate. While Hsp104 may not recognize all classes of unfolded proteins and aggregates, these results suggest that the transcriptional negative feedback loop is required to deactivate Hsf1 once proteostasis has been restored.

## Scanning mutagenesis reveals three independent repressive segments in Hsf1

In addition to positioning the transcriptional feedback loop as the core regulatory circuit that controls Hsf1 activity, the model also posits that Hsp70 binding is the mechanism that represses Hsf1. If this assumption were true, then disrupting the binding interaction should increase Hsf1 activity under non-heat shock conditions (*Figure 2—figure supplement 1*). To test this, we generated a series of 48 Hsf1 mutants in which we systematically removed 12 amino acid segments along the nonessential N- and C-terminal regions of Hsf1 (*Figure 2A*). We integrated these mutants into the genome as the only copy of *HSF1* in a strain background bearing the HSE-YFP reporter and assayed for activity by measuring YFP levels under non-heat shock and heat shock conditions by flow cytometry

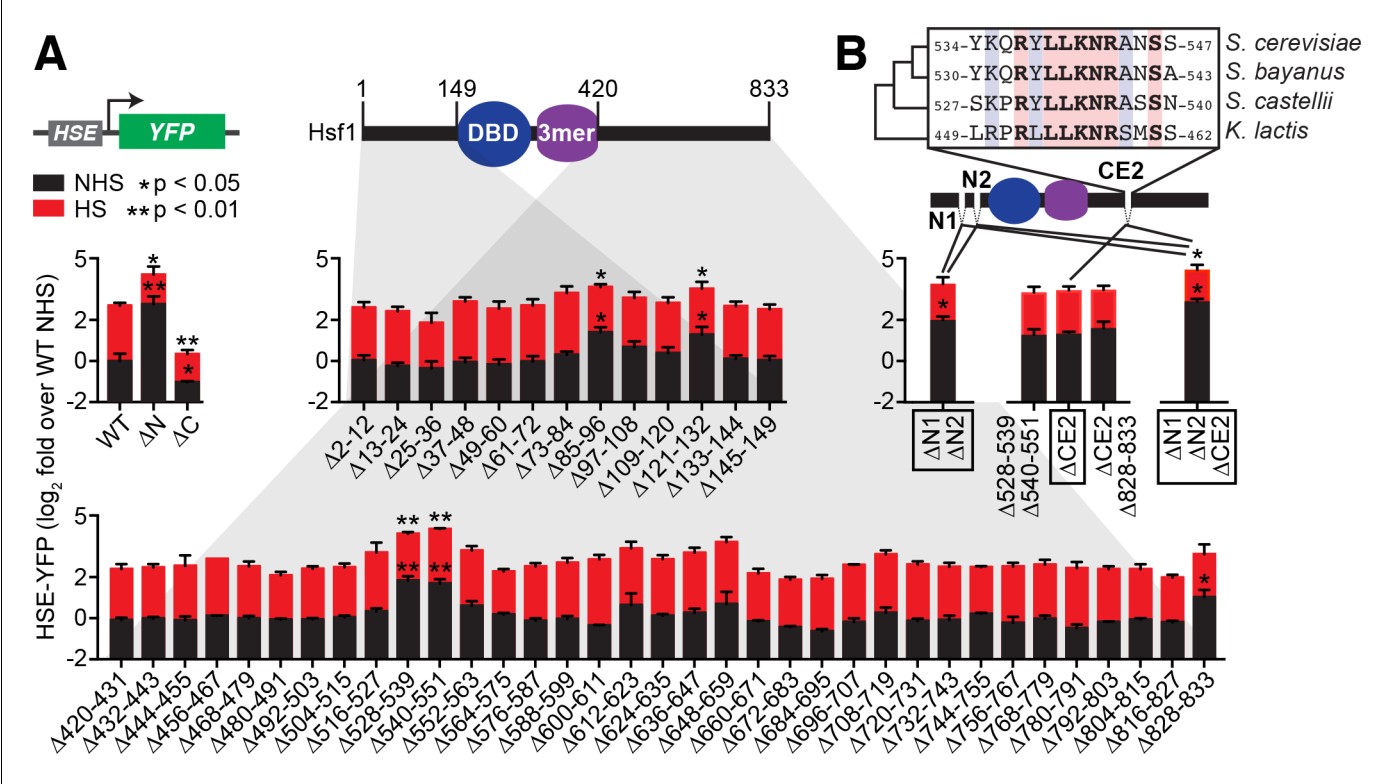

**Figure 2.** Identification of negative regulatory determinants in the N- and C-termini of Hsf1. (**A**) Screen for functional determinants. The indicated Hsf1 mutants were integrated into the genome as the only copy of Hsf1 expressed from the endogenous *HSF1* promoter in a strain expressing the HSE-YFP reporter. Hsf1ΔN is a deletion of the first 145 amino acids following the methionine; Hsf1ΔC is a truncation of the last 409 amino acids of Hsf1, retaining the first 424 amino acids. Each mutant in the scanning deletion analysis is missing a stretch of 12 amino acids in either the N-terminal 149 residues or final 414 C-terminal residues. Each strain was assayed in triplicate for its HSE-YFP level under non-heat shock (NHS) and heat shock (HS) conditions by flow cytometry. The error bars are the standard deviation of the replicates. Statistical significance was determined by one-way ANOVA (*p<0.05; **p<0.01). (**B**) Analysis of double and triple mutants of the functional segments. ΔN1 and ΔN2 represent Δ85–96 and Δ121–132, respectively, and each independently contribute to Hsf1 activity. CE2 is a region spanning the consecutive C-terminal determinants defined in (**A**) that is conserved among a subset of fungal species. Statistical significance was determined by one-way ANOVA comparing each double mutant to both of the single mutant parents (*p<0.05 for both tests).

DOI: https://doi.org/10.7554/eLife.31668.004

The following figure supplement is available for figure 2:

**Figure supplement 1.** Simulation showing an increase in the basal level of the HSE-YFP reporter as a function of increased dissociation rate (decreased affinity) of the Hsp70•Hsf1 interaction.

DOI: https://doi.org/10.7554/eLife.31668.005

(*Zheng et al., 2016*). To benchmark the assays, we used wild type Hsf1 and mutants lacking the entire N- and C-terminal regions. As previously shown, removal of the N-terminal region led to significantly increased Hsf1 activity under both non-heat shock and heat shock conditions in this assay (*Sorger, 1990*; *Zheng et al., 2016*), while removal of the C-terminal region significantly reduced Hsf1 activity under both conditions (*Figure 2A*). In the N-terminal region, we found two distinct 12 amino acid segments that when deleted resulted in increased Hsf1 activity (amino acids 85–96 and 121–132) (*Figure 2A*). In the C-terminal region, removal of two consecutive 12 amino acid segments as well as truncation of the final six amino acids resulted in increased Hsf1 activity (amino acids 528– 539, 540–551 and 828–833) (*Figure 2A*).

To determine if these segments acted independently, we generated double mutants. Combining the N-terminal deletions (Δ85–96/Δ121–132) resulted in a mutant with significantly greater basal activity than either of the single mutants, suggesting that these segments operate independently to repress Hsf1 activity (p<0.05, *Figure 2B*). We will refer to these N-terminal segments as N1 and N2. By contrast, combining the consecutive C-terminal segments (Δ528–539/Δ540–551) resulted in a

double mutant with the same activity as the single deletions, suggesting that a unique functional determinant encompasses these segments (*Figure 2B*). Consistent with this notion, a region spanning these two segments comprises a previously identified element conserved in Hsf1 in other fungal species known as 'conserved element 2' (CE2) (*Figure 2B*) (*Jakobsen and Pelham, 1991*; *Nicholls et al., 2011*). Indeed, specific removal of CE2 was sufficient to match the increased level of Hsf1 activity observed in the Δ528–539/Δ540–551 mutant (*Figure 2B*). Additional removal of the final six amino acids provided no further increase in Hsf1 activity, consistent with previous studies suggesting a non-additive interaction between these elements (*Figure 2B*) (*Hashikawa and Sakurai, 2004*; *Yamamoto et al., 2007*). However, combining the N1/N2 and CE2 deletions resulted in an Hsf1 mutant with significantly greater activity than either the ΔN1/ΔN2 mutant or the ΔCE2 mutant (*Figure 2B*). Together, the scanning mutagenesis revealed three independent repressive segments on Hsf1 (N1, N2, and CE2).

## The Hsf1 N-terminal region modulates DNA binding

The segments we identified with increased HSE-YFP levels could function either by enhancing the association of Hsf1 with HSEs (i.e., increasing DNA binding) or by boosting the transactivation capacity of Hsf1 (i.e., increasing recruitment of the transcriptional machinery). To directly test the ability to bind to HSEs in cells, we performed chromatin immunoprecipitation (ChIP) of wild type Hsf1, Hsf1$^{\Delta N}$, Hsf1$^{\Delta C}$, Hsf1$^{\Delta N1/\Delta N2}$, Hsf1$^{\Delta CE2}$ and Hsf1$^{\Delta N1/\Delta N2/\Delta CE2}$ under non-heat shock and acute (5 min) heat shock conditions. Following ChIP enrichment, we assayed for association with the synthetic *4xHSE* promoter that drives the YFP reporter as well as five endogenous target gene promoters (*HSC82, HSP82, SSA4, HSP26* and *TMA10*) by qPCR. Under non-heat shock conditions, wild type Hsf1 binding ranged over nearly two orders of magnitude across these targets, from 0.14% of input at the *TMA10* promoter to 12.0% of input at the *4xHSE* promoter (*Figure 3—figure supplement 1A*). Upon acute heat shock, the inducibility of Hsf1 binding also varied widely across these targets, with induction of greater than 100-fold for *TMA10* and less than 1.5-fold for *HSC82* (*Figure 3—figure supplement 1A*). These data are inconsistent with the notion that Hsf1 is constitutively bound to its target genes (*Gross et al., 1990*; *Jakobsen and Pelham, 1988*; *Sorger et al., 1987*).

Interestingly, the Hsf1$^{\Delta N}$ mutant showed significantly increased association with the *4xHSE* and *SSA4* promoters under non-heat shock conditions (*Figure 3A*, *Figure 3—figure supplement 1A*). This increased binding to the *4xHSE* promoter was accompanied by increased transcriptional output of the YFP reporter in Hsf1$^{\Delta N}$ cells (*Figure 3B*). Northern blot analysis of expression of the endogenous *SSA* transcripts corroborated the HSE-YFP results (*Figure 3—figure supplement 1B*). These data suggest a simple relationship between DNA binding and transcription for the Hsf1$^{\Delta N}$ mutant: the N-terminal region of Hsf1 inhibits DNA binding and thereby reduces transcriptional activity.

Consistent with a role for the N-terminal segment in regulating DNA binding, the Hsf1$^{\Delta N1/\Delta N2}$ mutant mirrored Hsf1$^{\Delta N}$ in both its increased binding to the *4xHSE* promoter and increased transcription of the YFP reporter under non-heat shock conditions relative to wild type (*Figure 3A,B*). However, unlike the complete ablation of the N-terminal region, Hsf1$^{\Delta N1/\Delta N2}$ showed no increase in association with the *SSA4* promoter compared to wild type (*Figure 3—figure supplement 1A*), suggesting that its enhanced association with endogenous targets may be limited. Neither Hsf1$^{\Delta CE2}$ nor Hsf1$^{\Delta N1/\Delta N2/\Delta CE2}$ showed significant differences compared to wild type at any of the six target promoters under either non-heat shock or heat shock conditions, indicating that CE2 has no effect on Hsf1 DNA binding (*Figure 3—figure supplement 1*). Remarkably, under heat shock conditions, none of the five mutants showed significant differences in binding to the *4xHSE* promoter compared to wild type (*Figure 3A*). Thus, during heat shock, the differences in YFP reporter levels reflect the different transactivation abilities of the series of mutants, spanning more than 16-fold between Hsf1$^{\Delta C}$ and Hsf1$^{\Delta N1/\Delta N2/\Delta CE2}$ (*Figure 3B*). Taken together, the ChIP results suggest that the N-terminal region inhibits DNA binding at select promoters with a major contribution from the N1 and N2 segments. This effect may be direct, reflecting an intrinsically higher affinity of Hsf1$^{\Delta N}$ or Hsf1$^{\Delta N1/\Delta N2}$ for DNA, or indirect, a consequence of enhanced ability to recruit chromatin remodeling enzymes to open local chromatin structure.

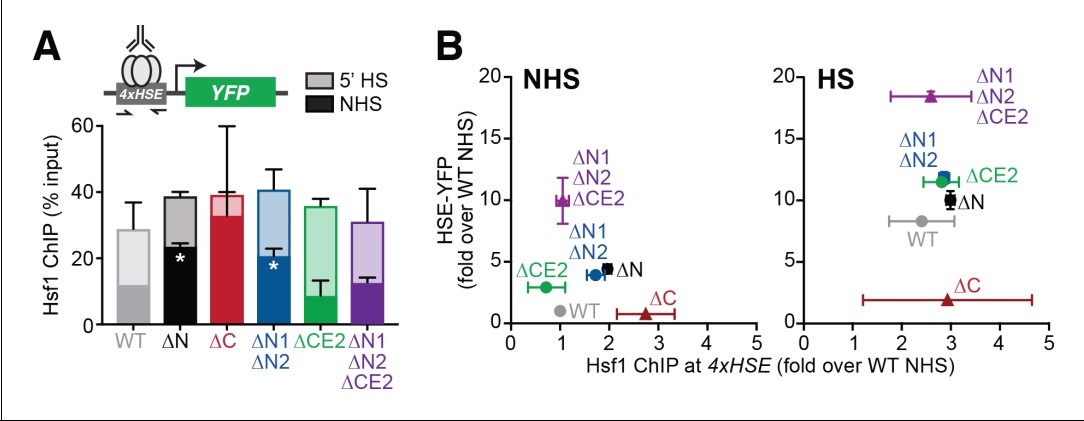

**Figure 3.** The Hsf1 N-terminus regulates DNA binding while CE2 controls transactivation. (**A**) Chromatin immunoprecipitation of Hsf1 followed by quantitative PCR of the *4xHSE* promoter in the indicated Hsf1 wild type and mutant strains under non-heat shock and heat shock conditions (solid and outlined bars, respectively). Error bars show the standard deviation of biological replicates. Statistical significance was determined by one-way ANOVA (*p<0.05; **p<0.01). (**B**) Relationship between Hsf1 binding at the 4xHSE promoter as determined by ChIP-qPCR and transcriptional activity as measured by levels of the HSE-YFP reporter under non-heat shock (NHS) and heat shock (HS) conditions for the panel of mutants assayed in (**A**).
DOI: https://doi.org/10.7554/eLife.31668.006

The following figure supplement is available for figure 3:

**Figure supplement 1.** Analysis of Hsf1 binding and induction of endogenous promoters and targets.
DOI: https://doi.org/10.7554/eLife.31668.007

## CE2 is necessary for Hsf1 to bind to Hsp70

Since CE2 affects Hsf1 transactivation but not DNA binding, we hypothesized that it could be a binding site for Hsp70. To test this, we performed serial immunoprecipitation from whole cell lysates followed by mass spectrometry (IP/MS) of 3xFLAG/V5-tagged Hsf1 mutants to identify specific interactions with chaperone proteins (*Zheng and Pincus, 2017*). We measured Hsp70 binding to wild type Hsf1, Hsf1$^{\Delta N}$, Hsf1$^{\Delta C}$, Hsf1$^{\Delta N1/\Delta N2}$, Hsf1$^{\Delta CE2}$ and Hsf1$^{\Delta N1/\Delta N2/\Delta CE2}$ under non-heat shock conditions, performing three biological replicates for each. Removal of the entire N-terminal region or the N1/N2 segments had no effect on Hsp70 binding relative to wild type (*Figure 4A*). By contrast, removal of the full C-terminal region significantly reduced the association of Hsf1 with Hsp70 (*Figure 4A*). Moreover, specific removal of CE2 – either alone or in combination with the N1/N2 deletions – also resulted in significantly diminished association with Hsp70, nearly matching removal of the entire C-terminal region (*Figure 4A*). Analysis of an additional biological replicate by Western blotting corroborated the IP/MS results (*Figure 4A*). The residual Hsp70 that co-precipitated with Hsf1$^{\Delta CE2}$ was refractory to dissociation upon heat shock, suggesting that this secondary interaction is unlikely to be regulatory (*Figure 4B*).

If CE2 is a direct binding site for Hsp70, then its primary sequence should control the affinity. Since Hsp70 binds best to peptides with hydrophobic and basic amino acids (*Van Durme et al., 2009*), we reasoned that mutation of three basic residues in CE2 to acidic residues should disrupt Hsp70 binding and lead to increased Hsf1 activity under non-heat shock conditions. Indeed, the Hsf1$^{R537E,K541D,R543E}$ triple mutant (referred to as ce2-mut) displayed increased levels of the HSE-YFP reporter under non-heat shock conditions, phenocopying ΔCE2 (*Figure 4C*). Conversely, introduction of two additional repeats of the wild type CE2 sequence (3xCE2) into Hsf1 reduced the basal level of the reporter (*Figure 4C*).

To test if CE2 is required for Hsf1 to directly bind to Hsp70, we utilized an in vitro binding assay we previously established to monitor interaction between recombinant purified Hsf1 and Hsp70 (*Zheng et al., 2016*). Whereas wild type Hsf1-6xHIS was able to outcompete wild type Hsf1-V5 for binding to Ssa2 (the most highly expressed yeast Hsp70) at 5-fold molar excess, Hsf1$^{\Delta CE2}$-6xHIS was not (*Figure 4D*, *Figure 4—figure supplement 1*). Together, these results demonstrate that CE2 is necessary for Hsp70 to bind to Hsf1 and repress its basal activity.

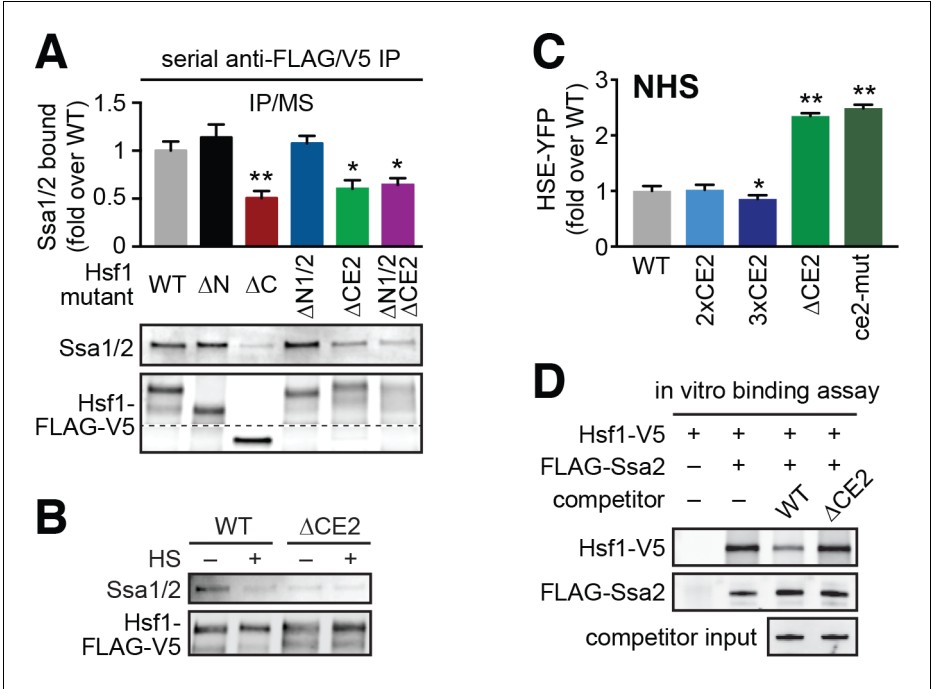

**Figure 4.** CE2 is necessary for Hsp70 to bind to Hsf1. (**A**) Co-immunoprecipitation of Hsf1 and Hsp70. The indicated Hsf1 mutants, C-terminally tagged with 3xFLAG-V5, were serially precipitated and subjected to mass spectrometry as described. The ratio of Hsp70 (Ssa1/2) to Hsf1 was determined in three biological replicates (bar graph, error bars are the standard deviation). Statistical significance was determined by one-way ANOVA (*p<0.05; **p<0.01). An additional replicate was analyzed by Western blot using antibodies against Ssa1/2 and the FLAG tag to recognize Hsf1. The FLAG blot was cropped in the middle to show the much smaller Hsf1$^{\Delta C}$. The immunoblot results are not as quantitative as MS and therefore were not used in generating bar graph. (**B**) Cells expressing C-terminally 3xFLAG-V5-tagged wild type Hsf1 and Hsf1$^{\Delta CE2}$ were either left untreated or heat shocked for 5 min at 39°C before serial Hsf1 immunoprecipitation and analyzed by Western blot using antibodies against Ssa1/2 and the FLAG tag to recognize Hsf1. (**C**) Cells expressing the indicated mutants of Hsf1, expressed as the only copy of Hsf1, were assayed for HSE-YFP levels under non-heat shock conditions by flow cytometry. The error bars are the standard deviation of three replicates. Statistical significance was determined by one-way ANOVA (*p<0.05; **p<0.01). (**D**) In vitro Hsf1:Hsp70 binding assay. Recombinant Hsf1-V5 and 3xFLAG-Ssa2 were purified, incubated together and assayed for binding by anti-FLAG immunoprecipitation followed by epitope-tag-specific Western blot. Addition of 5-fold molar excess of wild type Hsf1-6xHIS but not Hsf1$^{\Delta CE2}$-6xHIS diminished the amount of Hsf1-V5 bound to 3xFLAG-Ssa2.

DOI: https://doi.org/10.7554/eLife.31668.008

The following figure supplement is available for figure 4:

**Figure supplement 1.** Titration series of the in vitro Hsf1:Hsp70 binding competition assay.
DOI: https://doi.org/10.7554/eLife.31668.009

## CE2 is sufficient to bind to Hsp70

CE2 could be necessary for Hsf1 to bind to Hsp70 either because it is a direct binding site or because it influences the conformation of Hsp70 to expose a binding site located elsewhere. To test if CE2 is sufficient to bind to Hsp70, we developed an in vitro fluorescence polarization assay. We obtained synthetic peptides consisting of the wild type CE2 sequence or the ce2-mut sequence labeled at their N-termini with 5-carboxyfluorescein (5-FAM) (*Figure 5A*). Neither peptide aggregated in solution. We titrated the concentration of recombinant yeast Hsp70 (Ssa2) and measured polarization of the 5-FAM fluorophore. Using this assay, we determined that Ssa2 directly binds to CE2 with a dissociation constant ($K_d$) of 9 μM, while the $K_d$ for the ce2-mut peptide is reduced more than 5-fold (extrapolated to be 52 μM).

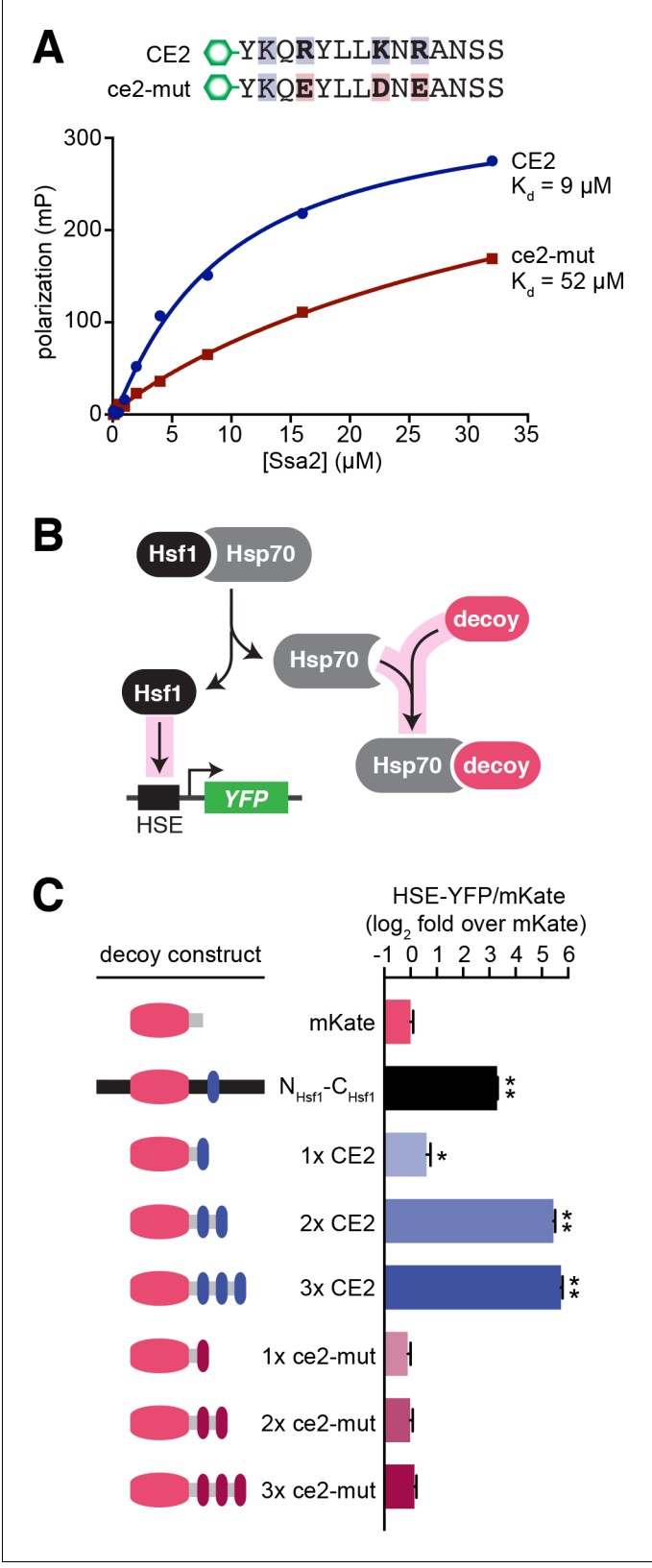

**Figure 5.** CE2 is sufficient to bind Hsp70. (**A**) Fluorescence polarization assay of CE2 and ce2-mut peptides labeled with 5-carboxyfluorescein (5-FAM) and recombinant Hsp70 (Ssa2). Peptides were maintained at 100 nM while Hsp70 was titrated at the indicated concentrations. Reactions were incubated for 30 min at room

*Figure 5 continued on next page*

*Figure 5 continued*

temperature prior to measurement. (**B**) Schematic of the 'decoy' assay. Overexpression of mKate-based decoy constructs activates Hsf1 in trans by titrating away Hsp70. (**C**) Decoy assay results. Cells bearing the indicated decoy constructs were induced with 1 µM estradiol for 16 hr at 30℃ and measured by flow cytometry. The HSE-YFP values in each cell were normalized by the expression of the decoy as measured by mKate fluorescence. Data are represented as median values of 10,000 cells relative to the median value of mKate alone. Error bars are the standard deviation of 3 biological replicates. Statistical significance was determined by one-way ANOVA (*$p<0.05$; **$p<0.01$).
DOI: https://doi.org/10.7554/eLife.31668.010

To test if the CE2 sequence is sufficient to bind to Hsp70 in cells, we deployed the 'decoy' assay we previously developed as a proxy to monitor Hsp70 binding (*Zheng et al., 2016*). In this experiment, we overexpressed synthetic constructs fused to a scaffold consisting of the well-folded fluorescent protein mKate that we can use to quantify expression. If the sequence appended to mKate binds to Hsp70, then it will titrate Hsp70 from endogenous Hsf1 and we will observe activation of the HSE-YFP reporter (*Figure 5B*). As a positive control, we used the previous decoy construct containing the full Hsf1 N- and C-termini ($N_{Hsf1}$-$C_{Hsf1}$), and we used mKate alone as a negative control. Addition of a single CE2 sequence to mKate modestly but significantly increased the HSE-YFP level, while addition of two or three tandem CE2 segments dramatically induced the HSE-YFP reporter, to a greater extent than even the $N_{Hsf1}$-$C_{Hsf1}$ decoy (*Figure 5C*). By contrast, no induction was observed for the ce2-mut decoys, even in the presence of three tandem repeats (*Figure 5C*). Thus, both in vitro and in cells, CE2 is sufficient to bind to Hsp70.

## Hsp70 affinity tunes the dynamics of the heat shock response

Finally, we returned to our mathematical model to predict the effects that modulating Hsp70:Hsf1 affinity would have on the dynamics of the heat shock response. Intuitively, simulations in which we increased the affinity showed faster deactivation kinetics and a lower maximal output, while decreasing the affinity showed slower deactivation and increased maximal output (*Figure 6A*, *Figure 6—figure supplement 1A*). To test these predictions experimentally, we utilized Hsf1$^{3xCE2}$ to increase affinity for Hsp70 and the Hsf1$^{ΔCE2}$ and Hsf1$^{ce2-mut}$ to reduce affinity. In agreement with the model, Hsf1$^{3xCE2}$ deactivated more rapidly than wild type, while Hsf1$^{ΔCE2}$ and Hsf1$^{ce2-mut}$ displayed delayed

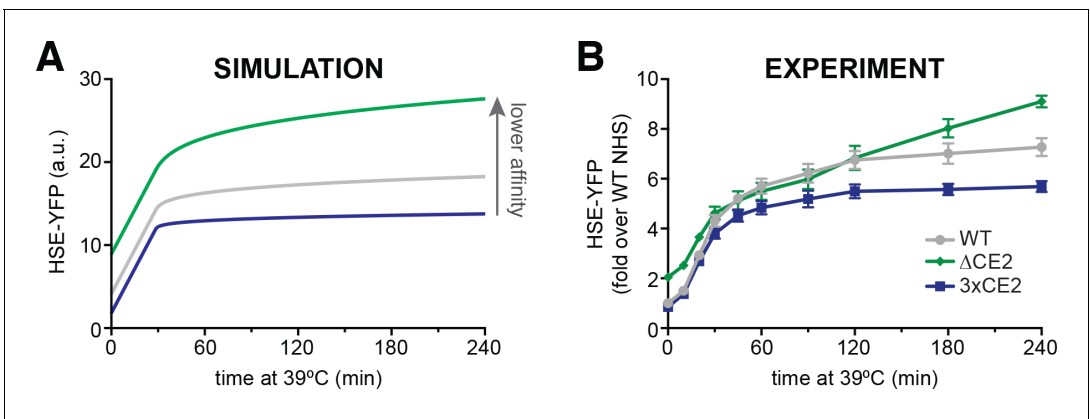

**Figure 6.** The affinity of Hsp70 for Hsf1 tunes the dynamics of the heat shock response. (**A**) Simulations of HSE-YFP levels over a heat shock time course as a function of increased rate of dissociation (reduced affinity) of Hsp70 from Hsf1. (**B**) Experimental heat shock time course of HSE-YFP levels in cells expressing wild type Hsf1, Hsf1$^{ΔCE2}$ or Hsf1$^{3xCE2}$. Each point represents the average of the median HSE-YFP level in three biological replicates, and the error bars are the standard deviation of the replicates.
DOI: https://doi.org/10.7554/eLife.31668.011

The following figure supplement is available for figure 6:

**Figure supplement 1.** Hsp70 affinity modulation alters Hsf1 activation kinetics.
DOI: https://doi.org/10.7554/eLife.31668.012

deactivation kinetics in a heat shock time course (*Figure 6B*, *Figure 6—figure supplement 1B*). Thus, the affinity for Hsp70 tunes Hsf1 activation dynamics.

## Discussion

In this study, we tested the assumptions of our mathematical model of the heat shock response by severing the Hsp70 transcriptional feedback loop and mapping an Hsp70 binding site on Hsf1. While we uncovered more biological complexity in Hsf1 regulation than we represent in the model, we validated the model's central tenets – that Hsp70 binding and dissociation turn Hsf1 off and on, and that transcriptional induction of Hsp70 represents a critical negative feedback loop required for the homeostatic regulation of Hsf1. Moreover, we found the model to be remarkably powerful in its ability to predict the dynamics of Hsf1 activity when challenged with targeted perturbations to the system architecture despite its oversimplified structure. These results argue that conceptualizing the heat shock response as a two-component feedback loop – in which Hsf1 positively regulates Hsp70 expression and Hsp70 negatively regulates Hsf1 activity – is an appropriate abstraction that captures the essence of the regulatory network. Whether this simplifying abstraction can be applied to HSF1 regulation in metazoans remains to be determined.

At a more mechanistic level, our screen for functional determinants in the N- and C-terminal regions of Hsf1 revealed three distinct segments in Hsf1 that independently exert negative regulation. The two N-terminal segments contribute to hitherto unknown repression of Hsf1 DNA binding, while the single C-terminal segment, CE2, is a binding site for Hsp70 through which Hsp70 represses Hsf1 transactivation. Although, as its name suggests, CE2 is conserved, it is restricted to a subset of yeast species and is absent in mammalian HSF1 sequences. Its amino acid composition, consisting of hydrophobic and basic residues, is reminiscent of peptide sequences known to bind to Hsp70 in vitro (*Van Durme et al., 2009*), though the affinity we measured (9 µM) is relatively weak. Given the stoichiometry of Hsp70:Hsf1 in cells (>500:1) (*Ghaemmaghami et al., 2003*) this weak affinity is likely necessary to allow for stress-dependent dissociation of the complex and has been a constant requirement of our mathematical model since its inception (*Zheng et al., 2016*). Thus, while CE2 is not conserved in mammalian genomes in primary sequence, it would seem facile to evolve a distinct but functionally analogous hydrophobic and basic segment to allow for weak Hsp70 binding. Notably, even though we found no evidence that the N1 segment is an additional Hsp70 binding site on endogenous Hsf1, its sequence is also predicted to be an Hsp70 binding site and is capable of binding to Hsp70 when overexpressed (S. Peffer and K. Morano, personal communication).

In addition to mechanistic insight into Hsp70 binding, our results for the first time reveal the existence of intramolecular determinants that negatively regulate Hsf1 DNA binding. While removal of the N-terminal region of Hsf1 leads to increased activity, at least in certain genomic contexts (*Sorger, 1990*) – suggesting that this region is repressive – the N-terminus also has a transactivation function and is important for efficient recruitment of Mediator during heat shock (*Kim and Gross, 2013*). Here we show that removal of the full N-terminal region results in increased association with select target gene promoters under non-heat shock conditions (*Figure 3A*), indicating a context-dependent role for this yeast-specific region in regulating DNA binding and suggesting a mechanistic basis for the increased transcriptional activity of Hsf1$^{\Delta N}$ relative to wild type Hsf1. In particular, the N1/N2 segments suppress DNA binding at the synthetic *4xHSE* promoter, as Hsf1$^{\Delta N/\Delta N2}$ displayed increased association (*Figure 3A*). If N1 were a bona fide second Hsp70 binding site (S. Peffer and K. Morano, personal communication), then this observation suggests that Hsp70 regulates both Hsf1 DNA binding and transactivation. Alternatively, if the N1/N2 segments impede DNA binding independent of Hsp70, then an additional heat shock-dependent mechanism would be required to relieve this block. Perhaps, by analogy to the intrinsic ability of human HSF1 to trimerize and bind DNA at elevated temperature (*Hentze et al., 2016*), the N1/N2 segments could contribute to direct thermosensing by mediating a temperature-dependent conformational change that increases DNA binding ability. Under either scenario, it is possible that the effect is indirect, and that deletion of N1/N2 results in the unmasking of the C-terminal activation domain.

Putting our observations together, we propose that Hsf1 can exist in one of four states in the yeast nucleus (*Figure 7*):

1. C-terminal activation domain (CTA) closed/DBD unbound

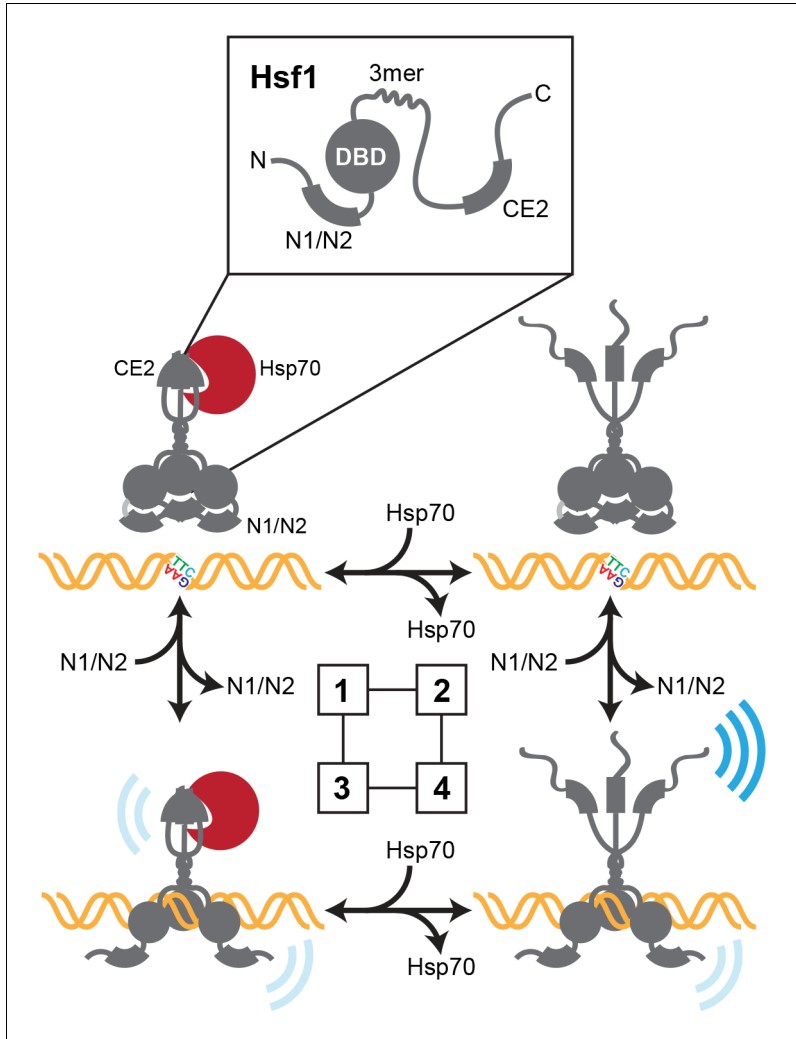

**Figure 7.** Thermodynamic representation of the four state model of Hsf1 activity. State 1: C-terminal activation domain (CTA) closed/DBD unbound Hsp70 is bound to CE2 keeping the CTA closed; the N-terminal region is engaged in blocking the DBD from accessing available HSEs via the N1/N2 segments. State 2 CTA open/DBD unbound: Hsp70 has dissociated from CE2; the CTA is open and can potentially recruit the transcriptional machinery; the N-terminal region continues to hinder DNA binding. State 3 CTA closed/DBD bound: Hsp70 remains bound to CE2 keeping the CTA closed; the N-terminal region has reoriented to allow HSE binding; Hsf1 weakly recruits the transcriptional machinery. State 4 CTA open/DBD bound: Hsp70 has dissociated from CE2 and the CTA is open; the N-terminal region has reoriented to allow HSE binding; Hsf1 avidly recruits the transcriptional machinery.

DOI: https://doi.org/10.7554/eLife.31668.013

 Hsp70 is bound to CE2 keeping the CTA closed; the N-terminal region is engaged in blocking the DBD from accessing available HSEs via the N1/N2 segments.

2. CTA open/DBD unbound
   Hsp70 has dissociated from CE2; the CTA is open and can potentially recruit the transcriptional machinery; the N-terminal region continues to hinder DNA binding.

3. CTA closed/DBD bound
   Hsp70 remains bound to CE2 keeping the CTA closed; the N-terminal region has reoriented to allow HSE binding; Hsf1 weakly recruits the transcriptional machinery.

4. CTA open/DBD bound
   Hsp70 has dissociated from CE2 and the CTA is open; the N-terminal region has reoriented to allow HSE binding; Hsf1 avidly recruits the transcriptional machinery.

The dual mechanisms of Hsf1 regulation described here – control of DNA binding and accessibility of the transactivation domain – in addition to the fine-tuning capacity we previously demonstrated for phosphorylation (*Zheng et al., 2016*), combine to exert exquisite quantitative control over the Hsf1 activity and the expression of its target gene regulon. We propose that these regulatory mechanisms enable cells to precisely tailor an optimal response to a variety of environmental and internal stresses.

## Materials and methods

### Yeast strains, plasmids and cell growth

Yeast cells were cultured in SDC media and dilution series spot assays were performed as described (*Zheng et al., 2016*). Strains and plasmids are listed in *Supplementary files 1* and *2*.

### Mathematical modleing

Modeling was performed as described (*Zheng et al., 2016*).

### Model parameter

| Parameter | Previous Paper model values | This paper's model values |
| --- | --- | --- |
| $k_1$, $k_3$ | 166.8 $min^{-1}$ $a.u.^{-1}$ | 166.8 $min^{-1}$ $a.u.^{-1}$ |
| $k_2$ | 2.783 $min^{-1}$ | 2.783 $min^{-1}$ |
| $k_4$ | 0.0464 $min^{-1}$ | 0.0464 $min^{-1}$ |
| $k_5$ | 4.64e-7 $min^{-1}$ | 4.64e-7 $min^{-1}$ |
| $\beta$ | 1.778 $min^{-1}$ | 0.3557 $min^{-1}$ |
| $K_d$ | 0.0022 a.u. | 0.0022 a.u. |
| $k_{dil}$ (fixed) | 0 $min^{-1}$ | 0 $min^{-1}$ |
| n (fixed) | 3 | 3 |

### Initial conditions

| Species | Initial value (a.u.) | Description |
| --- | --- | --- |
| $[HSP]_o$ | 1 | Free Hps70 |
| $[Hsf1]_o$ | 0 | Free Hsf1 |
| $[HSP•Hsf1]_o$ | 0.002 | HSP70•Hsf1 complex |
| $[HSP•UP]_o$ | 0 | Hsp70•UP complex |
| $[YFP]_o$ | 3 | Initial YFP concentration |
| $[UP]_o$ (@ 39°C) | 10.51 | UP concentration at 39°C |

### Flow cytometry

Heat shock experiments, heat shock time courses and decoy assays were performed and HSE-YFP levels were quantified by flow cytometry as described (*Zheng et al., 2016*). Data were processed in FlowJo 10. Data were left ungated and YFP fluorescence was normalized by side scatter (SSC) for each cell.

### Spinning disc confocal imaging

Imaging was performed as described (*Zheng et al., 2016*). Hsp104-mKate foci were quantified manually in ImageJ.

## Chromatin immunoprecipitation (ChIP)

Hsf1 ChIP was performed and quantified by qPCR as described (*Anandhakumar et al., 2016*).

## Serial 3xflag/V5 immunoprecipitation

Hsf1-3xFLAG-V5 was serially immunoprecipitated and analyzed by mass spectrometry and Western blotting as described (*Zheng et al., 2016*; *Zheng and Pincus, 2017*).

## Recombinant protein purification, binding and competition assay

Recombinant proteins were expressed and purified as described and the in vitro binding assay between Hsf1 and Ssa2 was performed as described (*Zheng et al., 2016*).

## Fluorescence polarization assay

CE2 and ce2-mut peptides labeled at their N-termini with 5-carboxyfluorescein (5-FAM) were obtained at >95% purity from GenScript. Fluorescence polarization was measured on a Tecan M1000 plate reader with absorbance at 480 nm and emission at 525 nm with increasing concentrations of 6x-HIS-3xFLAG-Ssa2. The peptides were kept constant at 100 nM. The reaction volume for each data point was 30 µl, and the measurements were performed in black, flat-bottomed 384 well plates after incubation for 30 min at room temperature. Binding curves were fitted using Prism software (Graph Pad) and $K_d$ values were extracted.

## Acknowledgements

We are grateful A Kane for providing us with the phleomycin resistance cassette and deleting *SSA3*, to A Jaeger for beneficial discussions, and to H Lodish, G Fink and their lab members for insightful comments. Experimentally, we are indebted to J Cheah at the Koch Institute Swanson Biotechnology Center High Throughput Screening facility for the fluorescence polarization measurements, to E Spooner and the Whitehead Proteomics core for mass spectrometric analysis, to the Whitehead Institute FACS facility for technical assistance and to N Azubuine and T Nanchung for a constant supply of plates and media.

## Additional information

### Funding

| Funder | Grant reference number | Author |
| --- | --- | --- |
| National Institutes of Health | DP5 OD017941-01 | David Pincus |
| National Science Foundation | MCB-1350949 | Ahmad S Khalil |
| National Science Foundation | MCB-1518345 | David S Gross |

The funders had no role in study design, data collection and interpretation, or the decision to submit the work for publication.

### Author contributions

Joanna Krakowiak, Xu Zheng, Zoë A Feder, Jayamani Anandhakumar, Investigation, Methodology; Nikit Patel, Formal analysis, Investigation, Methodology; Kendra Valerius, Investigation; David S Gross, Supervision, Funding acquisition, Methodology, Project administration, Writing—review and editing; Ahmad S Khalil, Conceptualization, Supervision, Funding acquisition, Visualization, Methodology, Project administration, Writing—review and editing; David Pincus, Conceptualization, Formal analysis, Supervision, Funding acquisition, Validation, Investigation, Visualization, Methodology, Writing—original draft, Project administration, Writing—review and editing

### Author ORCIDs

David Pincus ⓘD http://orcid.org/0000-0002-9651-6858

**Decision letter and Author response**
Decision letter https://doi.org/10.7554/eLife.31668.022
Author response https://doi.org/10.7554/eLife.31668.023

---

## Additional files

**Supplementary files**
• Source code 1. Simulation of Hsf1 activity with feedback in tact. MatLab code to simulate HSE-YFP reporter production with the Hsp70 feedback loop in tact.
DOI: https://doi.org/10.7554/eLife.31668.014

• Source code 2. Simulation of Hsf1 activity without feedback. MatLab code to simulate HSE-YFP reporter production with the Hsp70 feedback loop severed.
DOI: https://doi.org/10.7554/eLife.31668.015

• Source code 3. Simulation of basal Hsf1 activity as a function of its dissociation rate from Hsp70. MatLab code to simulate basal HSE-YFP levels as a function of the rate of dissociation of Hsp70 from Hsf1.
DOI: https://doi.org/10.7554/eLife.31668.016

• Source code 4. Simulation of dynamic Hsf1 activity as a function of its dissociation rate from Hsp70. MatLab code to simulate HSE-YFP levels over a heat shock time course as a function of the rate of dissociation of Hsp70 from Hsf1.
DOI: https://doi.org/10.7554/eLife.31668.017

• Supplementary file 1. Yeast strains used in this study. Excel table of plasmids.
DOI: https://doi.org/10.7554/eLife.31668.018

• Supplementary file 2. Yeast strains used in this study. Excel table of yeast strains with genotypes.
DOI: https://doi.org/10.7554/eLife.31668.019

• Transparent reporting form
DOI: https://doi.org/10.7554/eLife.31668.020

---

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
