## [Decision Letter]

Thank you for submitting your article "Hsf1 and Hsp70 constitute a two-component feedback loop that regulates the yeast heat shock response" for consideration by *eLife*. Your article has been reviewed by two peer reviewers, and the evaluation has been overseen by a Reviewing Editor and Jonathan Cooper as the Senior Editor. The following individuals involved in review of your submission have agreed to reveal their identity: Peter M Douglas (Reviewer #2).

The reviewers and the Reviewing Editor have drafted this decision to help you prepare a revised submission.

Here the authors have followed up their 2016 *eLife* paper where they reported that Hsp70/Ssa1-4 chaperone binding to the Hsf1 transcription factor was the main mechanism of negative regulation of Hsf1 in yeast, and that competition of Hsp70 away from Hsf1 in response to increased levels of unfolded protein induced by heat shock was the primary mechanism of activation of heat shock-induced Hsf1 target gene transcription. They also showed that the extensive phosphorylation of Hsf1 observed upon heat shock was dispensable for Hsf1 activation, but instead played a role in amplifying the Hsf1 transcriptional output. On this basis, they proposed that Hsf1 activity was regulated by two feedback loops – normally Hsp70 is in excess and can sequester Hsf1 by direct binding; following heat shock Hsf1 is released, and, among other target genes, drives expression of Hsp70, which then re-sequesters Hsf1 shutting off the response.

In this paper they tested their model in two ways. First, they engineered a yeast strain in which the four Ssa/Hsp70 genes were deleted and replaced by two copies of Ssa2 driven by a constitutive promoter, and showed that in this strain heat shock induced expression of an Hsf1-driven YFP gene with similar kinetics to WT, but that YFP expression was not shut off as heat shock persisted. Second, they showed that Hsp70 was able to bind to Hsf1 directly, and mapped the major binding site to a conserved element CE2 in the C-terminal tail of Hsf1.

The reviewers have discussed the reviews with one another and concur that this is a nice follow up to their prior paper, and is in principle appropriate for a Research Advance. However, although the data provide support for the model that the authors proposed in their earlier paper, the reviewers had a number of concerns that need to be addressed before publication

Essential points that need to be addressed:

1) The authors claim to have identified the binding site of Hsp70 on Hsf1. However, although their results indicate that CE2 is necessary for Hsp70 binding to Hsf1, they do not show that the CE2 motif is sufficient. For example, CE2 could influence the conformation of Hsf1 to expose an Hsp70 binding site that is located somewhere else, e.g. at the very C-terminal end of Hsf1, which also influenced the Hsf1 reporter activity. They should use a peptide-binding assay to demonstrate that this Hsf1 segment can be directly bound by Hsp70, and determine the Kd for binding this segment, and compare this value with the Kd of the Hsp70-Hsf1 complex used in their simulation. The authors should also create point mutations in this segment predicted to decrease the affinity of Hsp70 for Hsf1 more gradually, like replacing large hydrophobic residues (Leu or Tyr) for alanine or serine, or introducing negative charge in the CE2 region. These mutations should lead to predictable changes in the heat shock response that the authors could compare with simulations in their model.

2) Figure 1: It would be good to include (e.g. in the supplement) a chemical equation representation of the model with each arrow labeled with the appropriate parameter and indicating which parameters were changed in comparison to the previous version of the model. Although the model is described in the earlier published paper, the model is essential for the reader to understand this paper and inclusion of the model would facilitate understanding the study without continuous reference to the previous paper. Also, the authors mention that the original parameters did not represent their data correctly and that they had to change some parameters. It would be interesting to show a comparison of the simulation results in Figure 1 with the previous and the new parameter values side by side.

3) Figure 1: The key to the interpretation of the results is accurate quantification of the levels of Ssa/Hsp70 in the cells under different conditions. The authors used Pgk1 levels to normalize Ssa1/2 levels, but because its levels were increased in heat-shocked cells, Pgk1 cannot be used as a reliable loading control for quantification of Ssa1/2, and the authors need to use another protein whose levels do not change upon heat shock for quantification. In this regard, the quantification data need error bars, and significance analysis should be performed. Accurate Ssa/Hsp70 quantification is a critical issue, because as it stands the level of Ssa1/2 was apparently higher in the constitutive ΔFBL cells than in WT cells in the absence of heat shock (it is estimated as 1.6 fold higher). If this is the case and the sequestration model is correct, one might have expected the Hsf1-driven YFP response upon heat shock to be somewhat delayed in the constitutive strain. In addition, the Ssa2/Hsp70 level after heat shock induction in the WT strain was apparently not that much higher than in the constitutive strain without heat shock and yet this sufficient to shut off YFP expression – this raises the issue of whether the level of Hsf1, which was not measured, was the same in these two strains.

4) Figure 3: In the ChIP analysis, the ΔN1/ΔN2/ΔCE2 Hsf1 pulled down less HSE promoter DNA than the ΔN1/ΔN2 Hsf1 variant. According to the authors the N1 and the N2 region repress DNA binding in an Hsp70 independent manner, while CE2 contains the Hsp70 binding site for downregulation of the HSR. Why then does additional deletion of the Hsp70 binding site, i.e. prevention of Hsp70 binding, reduce DNA binding of the ∆N1/∆N2 Hsf1? Does Hsp70 association with to Hsf1 increase DNA binding? The ChIP data for those mutants are not consistent with the HSE-YFP reporter assay. With respect to DNA binding, the N1/N2 and CE2 elements seem to counteract each other, while with respect to Hsf1 transcriptional activity N1/N2 and CE2 act synergistically. Also, N1/N2 only affected the binding to the artificial 4xHSE promoter but not to any of the other promoters tested, and CE2 does not affect binding to any of the promoters in the context of the wild type Hsf1 protein. The question is whether either one of these elements affects expression of a reporter from a natural promoter. The authors should test their deletion variants using a reporter construct with a natural promoter like the Hsp82 and SSA4 promoters. In this regard, the fact that the ΔCE2 mutants resulted in direct upregulation of HSE-YFP activation, disagrees with previous work with *C. albicans* showing that CE2 is required for full expression of HSE genes (Nicholls et al., 2011), and this needs an explanation. In addition, the authors also fail to explain why individual deletions in the C-terminus cause increased YFP (but none caused decreased), while deleting the entire C-terminus results in a reduction in YFP.

5) Finally, it is not clear that this feedback mechanism applies to anything other than yeasts, since the CE2 element does not appear to be conserved in human HSF1, and this point merits at least some discussion.

---

## [Author Response]

Essential points that need to be addressed:1) The authors claim to have identified the binding site of Hsp70 on Hsf1. However, although their results indicate that CE2 is necessary for Hsp70 binding to Hsf1, they do not show that the CE2 motif is sufficient. For example, CE2 could influence the conformation of Hsf1 to expose an Hsp70 binding site that is located somewhere else, e.g. at the very C-terminal end of Hsf1, which also influenced the Hsf1 reporter activity. They should use a peptide-binding assay to demonstrate that this Hsf1 segment can be directly bound by Hsp70, and determine the Kd for binding this segment, and compare this value with the Kd of the Hsp70-Hsf1 complex used in their simulation.

We developed a fluorescence polarization assay using a labeled CE2 peptide and recombinant Ssa2 (Hsp70). As a control, we used a mutant CE2 peptide in which we replaced three basic residues with acidic residues (ce2-mut). We found the Kd of the Hsp70-CE2 complex to be 9 µM, a low but measurable affinity in line with the predictions of the mathematical model. The Hsp70-ce2mut complex had a Kd of 52 µM, indicating that the basic residues are important determinants of affinity and consistent with known Hsp70 binding preferences. In addition to this in vitro assay, we utilized the “decoy” assay we developed previously to show that appending the CE2 onto an inert mKate scaffold is sufficient to activate Hsf1 in trans in cells, with very strong activation if we increase the valency by appending two or three CE2 peptides. We have included these data in the revised manuscript (new Figure 5).

The authors should also create point mutations in this segment predicted to decrease the affinity of Hsp70 for Hsf1 more gradually, like replacing large hydrophobic residues (Leu or Tyr) for alanine or serine, or introducing negative charge in the CE2 region. These mutations should lead to predictable changes in the heat shock response that the authors could compare with simulations in their model.

We introduced several new CE2 mutations in full length Hsf1 and assayed their activity in cells. We replaced CE2 with the ce2-mut sequence (mutating the three basic residues to acidic residues) and increased the CE2 valency by adding two or three repeats of the CE2 sequence. Consistent with the rest of our results, ce2-mut phenocopied ∆CE2 in its increased basal activity, while the 3xCE2 mutant showed reduced basal activity and faster deactivation kinetics. We have included these results in the revised manuscript (Figure 4 and Figure 6).

2) Figure 1: It would be good to include (e.g. in the supplement) a chemical equation representation of the model with each arrow labeled with the appropriate parameter and indicating which parameters were changed in comparison to the previous version of the model. Although the model is described in the earlier published paper, the model is essential for the reader to understand this paper and inclusion of the model would facilitate understanding the study without continuous reference to the previous paper. Also, the authors mention that the original parameters did not represent their data correctly and that they had to change some parameters. It would be interesting to show a comparison of the simulation results in Figure 1 with the previous and the new parameter values side by side.

Only a single parameter changed in the model, and it is already highlighted in the Materials and methods in red in a table that directly compares the two versions of the model. We have added another supplemental figure as requested showing simulations of WT and the ∆FBL mutant with the original model parameter and the new parameter (Figure 1—figure supplement 1).

3) Figure 1: The key to the interpretation of the results is accurate quantification of the levels of Ssa/Hsp70 in the cells under different conditions. The authors used Pgk1 levels to normalize Ssa1/2 levels, but because its levels were increased in heat-shocked cells, Pgk1 cannot be used as a reliable loading control for quantification of Ssa1/2, and the authors need to use another protein whose levels do not change upon heat shock for quantification. In this regard, the quantification data need error bars, and significance analysis should be performed. Accurate Ssa/Hsp70 quantification is a critical issue, because as it stands the level of Ssa1/2 was apparently higher in the constitutive ΔFBL cells than in WT cells in the absence of heat shock (it is estimated as 1.6 fold higher). If this is the case and the sequestration model is correct, one might have expected the Hsf1-driven YFP response upon heat shock to be somewhat delayed in the constitutive strain. In addition, the Ssa2/Hsp70 level after heat shock induction in the WT strain was apparently not that much higher than in the constitutive strain without heat shock and yet this sufficient to shut off YFP expression – this raises the issue of whether the level of Hsf1, which was not measured, was the same in these two strains.

We repeated the western blot using GAPDH as a loading control, blotted for Hsf1 as well and included this new analysis in the revised manuscript (Figure 1). Hsf1 does not change its level in the ∆FBL strain, though there is more Ssa1/2 under non-heat shock conditions. The ratios displayed are the average of two biological replicates. The point of the ∆FBL strain is that Hsp70 cannot be induced by heat shock, and this is clearly shown.

4) Figure 3: In the ChIP analysis, the ΔN1/ΔN2/ΔCE2 Hsf1 pulled down less HSE promoter DNA than the ΔN1/ΔN2 Hsf1 variant. According to the authors the N1 and the N2 region repress DNA binding in an Hsp70 independent manner, while CE2 contains the Hsp70 binding site for downregulation of the HSR. Why then does additional deletion of the Hsp70 binding site, i.e. prevention of Hsp70 binding, reduce DNA binding of the ∆N1/∆N2 Hsf1? Does Hsp70 association with to Hsf1 increase DNA binding? The ChIP data for those mutants are not consistent with the HSE-YFP reporter assay. With respect to DNA binding, the N1/N2 and CE2 elements seem to counteract each other, while with respect to Hsf1 transcriptional activity N1/N2 and CE2 act synergistically.

Our ChIP results show that Hsf1∆CE2 pulled down less HSE promoter than WT; the ∆N1/∆N2 mutant pulled down more HSE promoter than WT; and the ∆N1/∆N2/∆CE2 pulled down the same amount as WT (the average of the ∆CE2 mutant and the ∆N1/∆N2 mutant). This suggests that these segments operate additively and independently. In the HSE-YFP assay, ∆N1/∆N2 increased activity, as did ∆CE2, and the ∆N1/∆N2/∆CE2 again was the sum of these two activities. There is no inconsistency here: in both the ChIP and HSE-YFP assays, the mutations are additive.

Also, N1/N2 only affected the binding to the artificial 4xHSE promoter but not to any of the other promoters tested, and CE2 does not affect binding to any of the promoters in the context of the wild type Hsf1 protein. The question is whether either one of these elements affects expression of a reporter from a natural promoter. The authors should test their deletion variants using a reporter construct with a natural promoter like the Hsp82 and SSA4 promoters.

Rather than generating new reporter constructs, we did a northern blot probing for endogenous Hsf1 target transcripts in the six strains under NHS and HS conditions. We used a probe that recognizes all four *SSA* transcripts. The expression levels were consistent with the reporter output, with the exception of the ∆C mutant (which is very hypoactive and quite sick). We attribute this discrepancy to the fact that the binding sites for the general stress response transcription factors Msn2/4 (STREs) in several of the *SSA* promoters, and the ∆C mutant likely has induced the general stress response to compensate for diminished Hsf1 activity. We included this northern blot in our revised manuscript (Figure 3—figure supplement 1).

In this regard, the fact that the ΔCE2 mutants resulted in direct upregulation of HSE-YFP activation, disagrees with previous work with C. albicans showing that CE2 is required for full expression of HSE genes (Nicholls et al., 2011), and this needs an explanation.

From our reading of the Nicholls et al. paper, they do not show the effect of target gene induction in the *C. albicans* ∆CE2 mutant at all. Rather, this is shown for the CE2t mutant, a truncation that includes removal of CE2 and the rest of the C-terminus. This CE2t mutant is more analogous to the ∆C mutant that we use here. They do show that the ∆CE2 mutant has a modest effect on growth at 37ºC, but this is just as likely to be due to hyperactivation as hypoactivation. We have now cited this paper, but there is no discrepancy.

In addition, the authors also fail to explain why individual deletions in the C-terminus cause increased YFP (but none caused decreased), while deleting the entire C-terminus results in a reduction in YFP.

We suspect that we failed to identify a single segment required for activation due to the nature of how the Hsf1 transcriptional activation domain works. The C-terminus is a ~400 amino acid intrinsically disordered region (IDR) that is required for heat shock-inducible transcriptional activity. It is likely that the association of multiple IDRs in Hsf1 trimers or higher order oligomers into subnuclear condensates is what recruits the transcriptional machinery and drives gene activation. Thus, deletion of any 12 aa segment is unlikely to disrupt the multi-valent IDR interactions.

5) Finally, it is not clear that this feedback mechanism applies to anything other than yeasts, since the CE2 element does not appear to be conserved in human HSF1, and this point merits at least some discussion.

We have expanded this part of the Discussion.